# Diving Deep into Regions: Exploiting Regional Information Transformer for Single Image Deraining

## Abstract

Transformer-based Single Image Deraining (SID) methods have achieved remarkable success, primarily attributed to their robust capability in capturing long-range interactions. However, we've noticed that current methods handle rain-affected and unaffected regions concurrently, overlooking the disparities between these areas, resulting in confusion between rain streaks and background parts, and inabilities to obtain effective interactions, ultimately resulting in suboptimal deraining outcomes. To address the above issue, we introduce the Region Transformer (Regformer), a novel SID method that underlines the importance of independently processing rain-affected and unaffected regions while considering their combined impact for high-quality image reconstruction. The crux of our method is the innovative Region Transformer Block (RTB), which integrates a Region Masked Attention (RMA) mechanism and a Mixed Gate Forward Block (MGFB). Our RTB is used for attention selection of rain-affected and unaffected regions and local modeling of mixed scales. The RMA generates attention maps tailored to these two regions and their interactions, enabling our model to capture comprehensive features essential for rain removal. To better recover high-frequency textures and capture more local details, we develop the MGFB as a compensation module to complete local mixed scale modeling. Extensive experiments demonstrate that our model reaches state-of-the-art performance, significantly improving the image deraining quality. Our code and trained models will be publicly available.

## 1 Introduction

Single Image Deraining (SID) remains a critical and persistently challenging task in the field of computer vision, primarily due to its extensive applications in fields such as surveillance Dey & Bhattacharjee (2020), autonomous driving Sun et al. (2019), and remote sensing Li et al. (2020). The inclusion of rain in images not only diminishes visual quality but also exerts a substantial influence on the effectiveness of numerous high-level vision tasks Chen et al. (2023a); Zhou et al. (2023). Eliminating rain streaks from rainy images poses an inherent challenge owing to the ill-posed nature of this problem, intricate patterns, occlusions, and diverse densities of raindrops, all of which collectively contribute to the complexity of the deraining process.

The emergence of deep learning has spearheaded significant progress in SID, with a multitude of Convolutional Neural Networks (CNN)-based methods being introduced Hu et al. (2019); Li et al. (2018); Yang et al. (2020). These methods have delivered promising outcomes across different scenarios. Beyond CNN-based approaches, Generative Adversarial Network (GAN)-based methods Qian et al. (2018); Zhang et al. (2019) have demonstrated their effectiveness in addressing the SID task. Through the use of adversarial loss, these models can produce visually appealing results, frequently surpassing traditional CNN-based techniques in preserving image details and structure. More recently, the remarkable success of transformer architecture in a broad range of computer vision tasks has prompted researchers to investigate its potential for SID Chen et al. (2023b); Jiang et al. (2022); Xiao et al. (2022).

Despite the substantial strides made with both transformer-based and CNN-based models in SID, when dealing with the complex and changeable real-world SID problem, especially when the rain streaks are very similar to the light spots in the image content, these methods often yield unsatisfactory

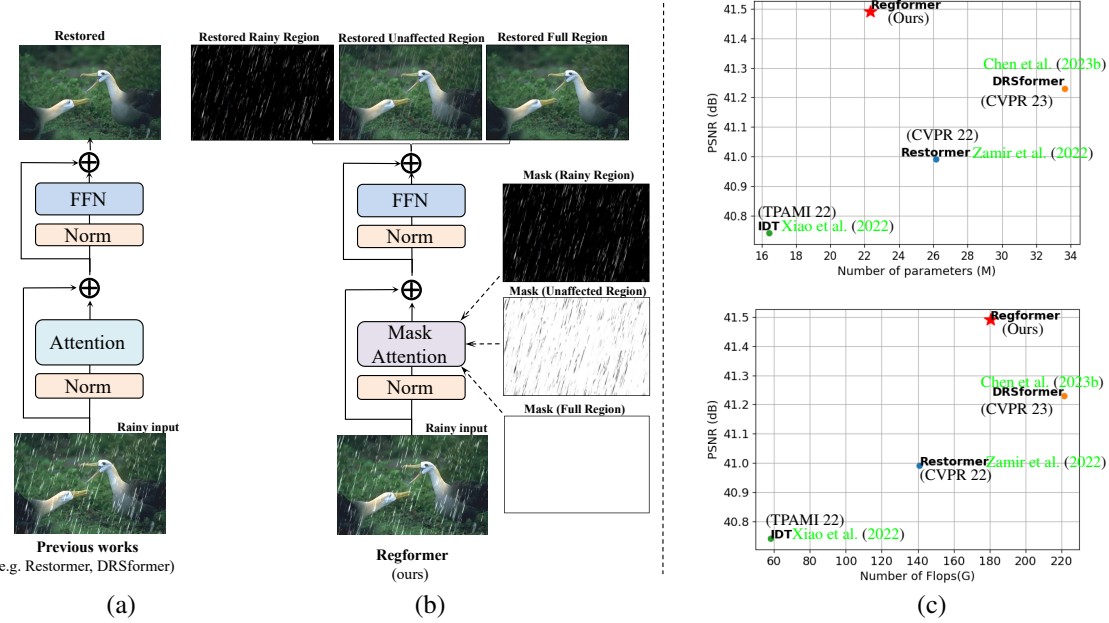

Figure 1: Sub-figures (a) and (b): Schematic diagrams illustrating the process of handling images in previous methods (left) and our Regformer (right). Different from previous works, our method applies three distinct masks to the attention phase, resulting in three different attentional results for SID. Our approach addresses the significant disparities between the rain-affected and unaffected regions of an image, as demonstrated in the output results of our approach. Sub-figure (c): Performance comparison between our method and other Transformer-based methods in terms of PSNR, computational complexity (GFLOPs), and model parameters. In both graphs, our method outperforms the others by achieving the highest PSNR while maintaining a reasonable balance of model complexity and computational cost. This showcases the efficiency and effectiveness of our Regformer for SID.

results, as can be seen from Fig 2. Upon a closer examination, it can be observed that current models do not differentiate between the rain-affected and unaffected during processing. However, there are significant disparities between the unaffected regions of an image and the rain-affected areas, so it's crucial to exploit the unique feature information inherent in these separate regions. Therefore, we believe that the distinct processing requirements of rain-affected and unaffected regions in images necessitate a more nuanced treatment than provided by current methods, as shown in Figure 1(a) and (b).

Drawing inspiration from these insights, we introduce a novel SID approach called **Reg**ion Trans**former** (Regformer). Regformer is meticulously designed to process rain-affected and unaffected regions separately while simultaneously considering their interaction to enhance image reconstruction. The cornerstone of our Regformer model lies in the innovative Region Transformer Block (RTB), which incorporates a Region Masked Attention (RMA) mechanism. RMA applies selective masking to the query (Q) and key (K) values during the attention process, thus generating attention feature maps exclusive to both the rain-affected and unaffected image areas. This design empowers our Regformer model to efficiently capture a more comprehensive set of features essential for efficient deraining while minimizing interference from extraneous or irrelevant features.

In addition, we introduce a novel module named Mixed Gate Forward Block (MGFB) to refine and enhance our model's feature extraction capabilities. MGFB is expertly designed to process features across varying receptive fields. This technique models local interactions more effectively, thereby enabling the model to gain a comprehensive understanding of the image context by considering multiple-scale local features. With the embedded Mixed-Scale Modeling process, MGFB can better recover textures and preserve local details. The outcome is a highly refined feature set, resulting in more robust and detailed representations. In conjunction, the RTB with its RMA mechanism and the MGFB module empowers our Regformer model to overcome the constraints faced by existing methods, further leading to state-of-the-art performance, as shown in Figure 1(c).

In summary, the contributions of our work are three-fold.

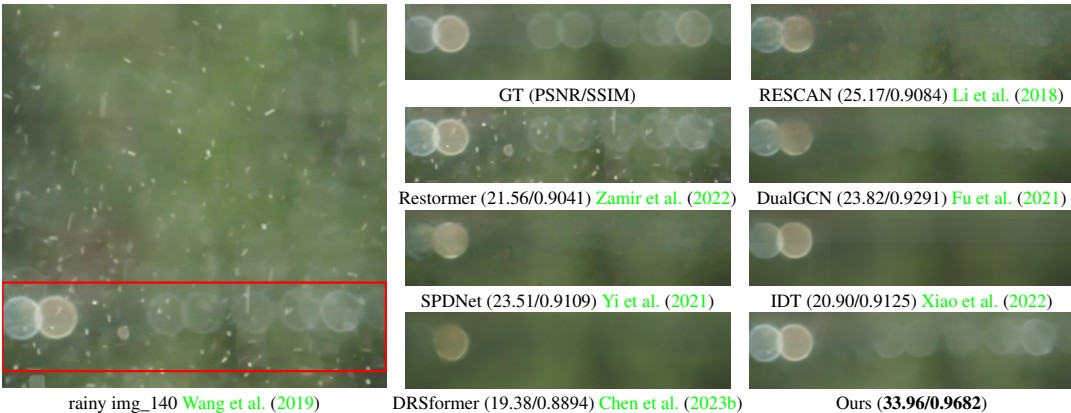

GT (PSNR/SSIM)    RESCAN (25.17/0.9084) Li et al. (2018)

Restormer (21.56/0.9041) Zamir et al. (2022)    DualGCN (23.82/0.9291) Fu et al. (2021)

SPDNet (23.51/0.9109) Yi et al. (2021)    IDT (20.90/0.9125) Xiao et al. (2022)

rainy img_140 Wang et al. (2019)    DRSformer (19.38/0.8894) Chen et al. (2023b)    Ours (**33.96/0.9682**)

Figure 2: Visual comparison of different methods on SPA-Data Wang et al. (2019) dataset. Clearly, our method can better preserve the original unaffected details of the input image compared to other methods.

- We present the innovative Regformer model, which introduces a novel region attention mechanism within the transformer to acquire the essential features for SOTA SID.

- We propose the innovative RTB with the RMA mechanism, enabling the model to efficiently capture comprehensive features from rain-affected, unaffected image areas, and their interactions, which are essential for effective deraining. Besides, the newly designed MGFB can perceive the local mixed-scale features between adjacent pixels in the image to complete local mixed-scale modeling.

- Our model achieves state-of-the-art performance on six public datasets, including four synthetic datasets and two real-world datasets. Our pretrained models and code will be publicly available.

## 2 RELATED WORK

In this section, we provide a comprehensive review of the literature on image deraining, specifically focusing on rain streak removal and raindrop removal techniques. Additionally, we discuss the development of Vision Transformers (ViT) and their applications in various vision tasks.

### 2.1 RAIN STREAK AND RAINDROP REMOVAL

In recent years, deep learning-based approaches have outperformed their conventional counterparts in rain streak removal. These methods employ semi-supervised methods Wei et al. (2019), self-supervised module Huang et al. (2021), patch-based priors for both the unaffected and rain layers Huang et al. (2022), recurrent squeeze-and-excitation contextual dilated networks Li et al. (2018), conditional GAN-based models Zhang et al. (2019), and two-step rain removal techniques Ahn et al. (2022); Lin et al. (2020) to achieve promising results. As a unique rain degradation phenomenon, raindrop removal presents unique challenges that most rain streak removal algorithms cannot address directly. Consequently, researchers have proposed dedicated techniques for raindrop detection and removal, e.g., attentive GAN-based networks Qian et al. (2018); Zhang et al. (2019); Wei et al. (2021), CNN-based networks with shape-driven attention Quan et al. (2019), and two-stage video-based raindrop removal approaches Chen et al. (2022); Yan et al. (2022).

### 2.2 VISION TRANSFORMERS IN IMAGE DERAINING

Following the Transformer network Vaswani et al. (2017), researchers have endeavored to replace CNNs with Transformers as the backbone structure for vision tasks. Dosovitskiy *et al*. Dosovitskiy et al. (2021) introduced the ViT by dividing an image into non-overlapping small patches and treating each patch as a token in the original transformer. Liu *et al*. Liu et al. (2021) proposed the Swin Transformer to further reduce the computational complexity of ViT. The Swin Transformer employs multi-headed self-attention in local windows rather than the global approach used in ViT. This modification reduces computation cost, while the shifted window approach compensates for any loss of global information.

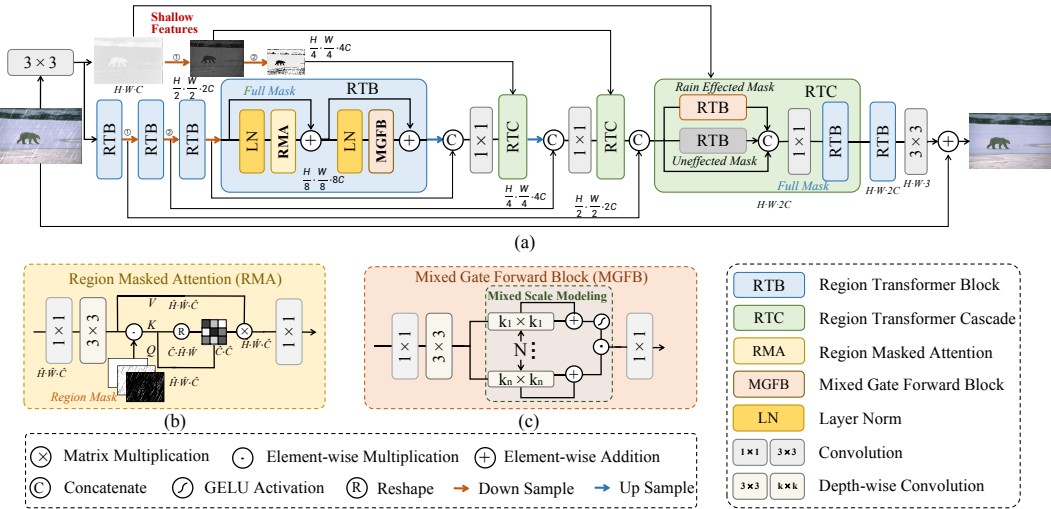

Figure 3: This is an overview of our proposed Regformer, which is comprised of Region Transformer Block (RTB) and Region Transformer Cascade (RTC) consisting of three RTBs (each implementing a different mask mechanism). In sub-figure (a), RTB with different colors represent the use of different mask strategy, and we use full mask strategy in the encoder part. We calculate the mask by utilizing the shallow features derived from the input image post a $3 \times 3$ convolution and various downsampling stages, along with the features obtained from the RTC (more details can be seen in Eq. 2 and Fig. 4). Sub-figure (a) depicts the overall framework of our model, sub-figure (b) demonstrates our RMA mechanism, and sub-figure (c) illustrates our MGFB mechanism.

In the domain of image rain removal, several notable contributions have been made. For instance, Jiang *et al*. Jiang et al. (2022) proposed a dynamic-associated deraining network that integrates self-attention into a Transformer alongside an unaffected recovery network. Following this, Xiao *et al*. Xiao et al. (2022) introduced the Image Deraining Transformer (IDT), which employs both window-based and spatial-based dual Transformers, achieving remarkable results. Further advancements were made by Zamir *et al*. Zamir et al. (2022), who presented an efficient transformer model named Restormer. By incorporating innovative designs in the building blocks, Restormer can capture long-range pixel interactions while maintaining applicability to large images, which significantly contributes to subsequent image restoration. Building upon the foundation laid by Restormer, Chen *et al*. Chen et al. (2023b) put forward a sparse transformer. This model further explores useful self-attention values for feature aggregation and introduces a learnable top-k selection operator to procure more detailed features. Despite these efforts, current methodologies often stumble in comprehensively addressing the distinct regions within an image and their interactions, curtailing image deraining quality, as depicted in Figure 2. These limitations serve as the impetus for our Regformer model.

## 3 METHOD

In this section, we present Region Transformer (Regformer) model for SID. First, we briefly introduce the overall architecture of the Regformer, followed by a detailed description of the critical components, i.e., the Region Transformer Block (RTB) with the Region Masked Attention (RMA) mechanism, and the Mixed Gate Forward Block (MGFB).

### 3.1 OVERALL ARCHITECTURE

The overall pipeline of our Regformer model, as shown in Figure 3(a), employs an encoder-decoder architecture. Given an input image of size $H \times W \times C$, where $H$, $W$, and $C$ represent the height, width, and channel number of the image, respectively, we first preprocess the image using a $3 \times 3$ convolution to extract the shallow feature. Note that we save a feature map during this stage, which is later utilized in the decoder part to obtain the mask. Subsequently, the feature map is passed through the RTB to extract global image features. Following each RTB in the encoder, we perform downsampling using $1 \times 1$ convolutions and pixel-unshuffle operations, similar to methods Chen et al. (2023b); Wang et al. (2022); Xiao et al. (2022); Zamir et al. (2022). Concurrently, we introduce skip connections to bridge intermediate features and maintain network stability. Each RTB comprises one

RMA, one MGFB, and two normalization layers. *Note that we directly use a tensor with all 1 values as a mask at the encoder stage, since we have not yet accessed the restored feature maps.*

In the decoder, we use $1 \times 1$ convolution to reduce channels after processing the skip connections. Our Region Transformer Cascade (RTC) in the decoder consists of three RTBs and a $1 \times 1$ convolution. Specifically, we duplicate the input feature maps and feed them into two RTBs separately. Within these two RTBs, we perform rain-affected and unaffected attention masking, resulting in two feature maps corresponding to the rain-affected and unaffected regions. A residual connection is then applied to avoid feature loss. We concatenate this resulting matrix and pass it through a $1 \times 1$ convolution to reduce the channels. Finally, an RTB without a masking mechanism is used to integrate the three regional feature maps, yielding a comprehensive map containing rich feature information from the three distinct regions. The above process can be expressed as:

$$F_{final} = RTB(Conv_{1 \times 1}(Concat(RTB_U(I), RTB_R(I), I))), \tag{1}$$

where $I$ denotes the input feature map, $RTB_U(\cdot)$ and $RTB_R(\cdot)$ represent the RTB process with unaffected region mask and rain region mask, respectively. The $Concat(\cdot)$ means using concatenation, and $Conv_{1 \times 1}(\cdot)$ represents using $1 \times 1$ convolution. In decoders with different depths, we process masks to varying extents. For the third layer decoder, we directly send the feature map extracted and saved from the encoder. For the first and second layer decoders, we need to downsample the feature map. Notably, the downsampling process shares parameters with the first two downsampling operations in the encoder part, as shown in Figure 3 with shared parameters indicated by the same number.

In the final stage, we use an RTB without masked regions to synthesize features and reconstruct the image, followed by a $3 \times 3$ convolution to adjust the channels. We then add a residual connection between the original input image and the output to preserve features during network processing.

## 3.2 REGION TRANSFORMER BLOCK

The Region Transformer Block (RTB) is composed of one RMA, one MGFB and two LN layers, as depicted in Figure 3(a). Specifically, we first pass the input feature map through the LayerNorm layer. Subsequently, we obtain the attention weight feature map using our RMA mechanism and apply a residual connection to preserve features. Following this, we perform normalization processing and comprehensive extraction of regional features through our MGFB module. The details of RMA and MGFB will be elaborated in the following subsections.

### 3.2.1 REGION MASKED ATTENTION

Previous works did not fully consider the unique characteristics of different regions within an image. This oversight often resulted in a lack of differentiation between the rain-affected and unaffected areas, which in turn limited the effectiveness of their deraining outcomes. To address the limitation of previous works, we propose region masked attention, as illustrated in Figure 3(b). We first pass the input feature map through a 1x1 convolution to obtain the query (Q), key (K), and value (V) matrices. We then apply a $3 \times 3$ depth-wise separable convolution as a preprocessing step. After getting the processed result, we divide Q, K, and V to generate three independent projection matrices of key-value pairs. It is essential to use masking to Q and K after this step. **Embedded Region Mask.** At different stages of our Regformer model, we employ various masking strategies to effectively differentiate between the rain-affected and unaffected regions of the input feature maps. As mentioned earlier, these masking approaches are tailored to suit the specific needs of different positions within the network architecture. The mask acquisition method is illustrated in Figure 4. We denote the feature

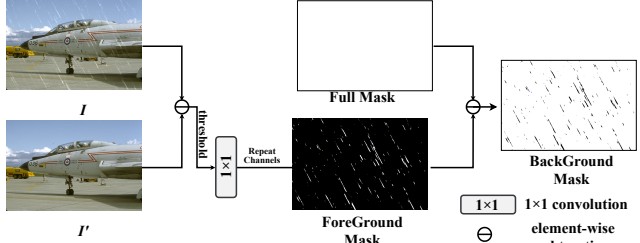

Figure 4: Illustration of the Region Mask generation. Here, $I$ and $I'$ represent the shallow features and the restored features in RTC, respectively. The ForeGround Mask focus on rain-affected region, and the BackGround Mask highlights the unaffected region. For ease of understanding , we show images mapped to RGB space rather than feature maps.

map after the $3\times3$ convolution as $I$ and downsample the feature map output, which is saved in the encoder part, to different degrees. In the decoder part, we denote the feature map before entering the next RTC as $I'$. Our rain mask (ForeGround Mask) $R$ and unaffected region mask (BackGround Mask) $U$ can be obtained using the following formula:

$$\begin{cases} R = Binarize(T(I - I')) * Conv_{1\times1}(Rep_C(1)), \\ U = 1 - R, \end{cases} \tag{2}$$

where $I$ and $I'$ denote the shallow features and the restored features in RTC, respectively; $R$ and $U$ denote the rain masks and unaffected region masks, respectively; $T$ signifies the application of dynamic thresholds; the $Binarize(\cdot)$ performs binarization, and $Rep_C(1)$ indicates the process of copying a single channel into C channels to align them. We use a dynamic threshold, to allow the network to adaptively distinguish the rain region and the unaffected region, and we binarize the segmentation result into a matrix of 0 or 1. In this case, the area where the value is set to 0 is not easily selected when processing the attention mechanism. However, this threshold cannot completely distinguish the rain region and unaffected region, which is shown in figure 6 and 7, so the RTB with the background mask is also needed. As a result, we derive the desired masks for rain-affected and unaffected regions.

**Region Masked Attention.** We then perform element-wise multiplication of the $Q$ and $K$ key-value pairs with the corresponding region mask to obtain the masked $Q$ and $K$ key-value pairs. Subsequently, we reshape $K$ to enable matrix multiplication, resulting in a region mask matrix with dimensions equal to the number of channels ($\hat{C}$). By performing matrix multiplication with this matrix and the $V$ key-value pair, we obtain the reconstructed attention map. The above process can be expressed as:

$$\begin{cases} Q' = Q \otimes M, \\ K' = Reshape(K \otimes M), \\ Attention(Q, K, V, M) = Conv_{1\times1}(Q'K' \otimes V), \end{cases} \tag{3}$$

where $Q$, $K$, and $V$ represent the query, key, and value, respectively. The $M$ represents the Region Mask mechanism, and the $\otimes$ here represents the element-wise multiplication. Hence, RMA effectively utilizes the rain mask and unaffected region mask, enabling our model to selectively focus on crucial information from both regions. Consequently, it ensures the preservation of critical information from both the rain-affected and unaffected regions while considering their combined perspective for high-quality image reconstruction, ultimately yielding a feature map with region-specific features. Finally, we use one $1\times1$ convolution to adjust the feature map's channels.

### 3.2.2 MIXED GATE FORWARD BLOCK

In the field of SID, one key point is how to conduct broader modeling for rain streaks of different scales. However, existing methods have not paid much attention to this Zamir et al. (2022), or are only limited to a specific scale modeling Chen et al. (2023b); Xiao et al. (2022), which will make the model perform poorly when retaining rain-like features, as can be seen from figure 2. In order to solve this problem, we design the Mixed Gate Forward Block (MGFB) with a Mixed Scale Modeling process to effectively perceive the local mixed-scale features in adjacent pixel areas, as depicted in Figure 3(c). Specifically, the input fused feature map is first passed through a $1\times1$ convolution to expand the channels, and we use a $3\times3$ depth-wise convolution to complete the preliminary feature extraction. This can be described as:

$$M = DWConv_{3\times3}(Conv_{1\times1}(I)), \tag{4}$$

where $I$ and $M$ represent the input feature map and the middle feature map, respectively. To complete Mixed Scale Modeling, we divide this feature map into multiple parts according to the channels and use depth-wise convolutions with different kernel sizes to fully extract features on different receptive fields. We added residual connections to avoid the loss of original features. The depth-wise convolutions with different kernels are used to capture the local features of rain streaks at varying scales without incurring a high computational cost, allowing the image deraining process to be guided simultaneously in different scales. Subsequently, the resulting feature maps are combined through element-wise matrix addition. This approach enables each level to focus on unique, detailed aspects complementary to other levels. This Mixed Scale Modeling process can be expressed as:

$$F = Activation(DWConv_{k_1\times k_1}(M) + M) \odot \prod_{i=2}^{n}(DWConv_{k_i\times k_i}(M) + M), \tag{5}$$

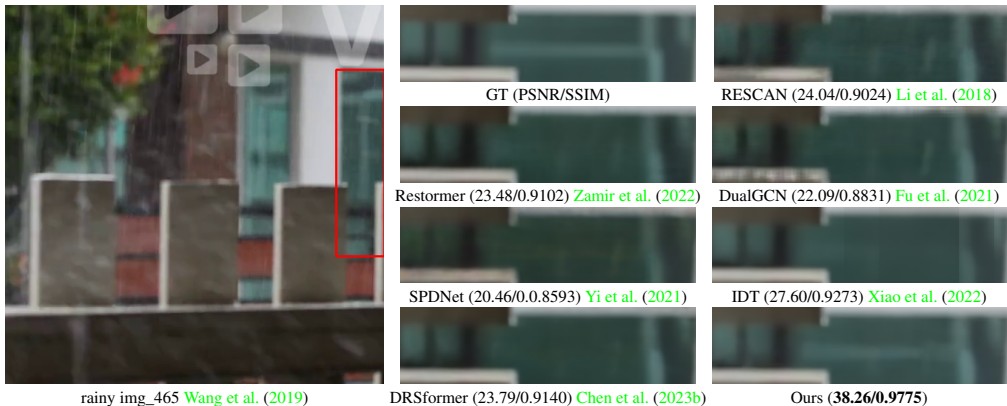

Figure 5: Visual comparison on SPA-Data Wang et al. (2019) dataset. To facilitate the display of the results, we rotate the local graph 90 degrees clockwise. Unlike previous methods, our method successfully preserves the original details of the input image, compared to the GT.

where $F$ denotes the final feature map, $DWConv(\cdot)$ represents using depth-wise convolution, the *Activation* refers to the activation function, the product symbol and $\odot$ here represents the element-wise multiplication. The $n$ and $k_i$ here are all hyperparameters. Lastly, we use a $1 \times 1$ convolution to adjust the number of output channels, which can be expressed as follows:

$$O = Conv_{1 \times 1}(F), \tag{6}$$

where $O$ represents the output feature map, and $Conv_{1 \times 1}$ represents using $1 \times 1$ convolution. By incorporating the MGFB, our model effectively processes the three-region features extracted by the RMA, while completing local mixed-scale modeling and resulting in a more comprehensive understanding of the image context and improved image deraining performance.

## 4 EXPERIMENTS

### 4.1 EXPERIMENTAL SETTINGS

**Evaluated Datasets.** We evaluate the performance of our model on several public benchmarks, including Rain200L Yang et al. (2017), Rain200H Yang et al. (2017), DID-Data Zhang & Patel (2018), DDN-Data Fu et al. (2017) and SPA-Data Wang et al. (2019). Specifically, Rain200L and Rain200H consist of 1,800 synthetic rainy image pairs for training and 200 synthetic test image pairs. DID-Data and DDN-Data contain 12,000 and 12,600 synthetic rainy image pairs for training, respectively, along with 1,200 and 1,400 rainy image pairs for testing. Meanwhile, SPA-Data is a real-world dataset that includes 638,492 training image pairs and 1,000 testing image pairs. We further evaluate Regformer on a real-world raindrop dataset AGAN-Data Qian et al. (2018), which includes 1,110 image pairs.

**Compared Methods.** For rain streak removal, we compare our Regformer model with a variety of state-of-the-art methods, including prior knowledge-based approaches like DSC Luo et al. (2015) and GMM Li et al. (2016), CNN-based methods such as DDN Fu et al. (2017), RESCAN Li et al. (2018), PReNet Ren et al. (2019), MSPFN Jiang et al. (2020), RCDNet Wang et al. (2020), MPRNet Zamir et al. (2021), DualGCN Fu et al. (2021) and SPDNet Yi et al. (2021). In addition, we consider Transformer-based methods, including Uformer Wang et al. (2022), Restormer Zamir et al. (2022), IDT Xiao et al. (2022) and DRSformer Chen et al. (2023b). We refer to the experimental data provided by DRSformer and adhere to their original experimental settings for further analysis. Our experimental results are presented in Table 1. In addition, we also test our model on one real-world raindrop dataset AGAN-Data Qian et al. (2018) to measure our model performance more broadly. We compare our model with five methods, including Eigen *et al.* Eigen et al. (2013), Pix2Pix Isola et al. (2017), AttentGAN Qian et al. (2018), Quan *et al.* Quan et al. (2019) and IDT Xiao et al. (2022). Our experimental results are presented in Table 2. We employ PSNR (Peak Signal-to-Noise Ratio) Q. & M. (2008) and SSIM (Structural Similarity Index Measure) Wang et al. (2004) as the primary evaluation metrics. Note that we calculate the metrics on the Y channel of YCbCr like the previous works Chen et al. (2023b); Wang et al. (2020); Yang et al. (2017).

**Implementation and Training Details.** In our Regformer model, we configure the layers L0, L1, L2 and L3 to 4, 4, 4, 4, and the corresponding attention heads to 6, 6, 6, 6. Additionally, we set the

Table 1: Quantitative comparison(PSNR/SSIM) with state-of-the-art methods for single image rain streak removal. The best and second best results are colored with red and blue.

| Datasets | Rain200L | | Rain200H | | DID-Data | | DDN-Data | | SPA-Data | |
|---|---|---|---|---|---|---|---|---|---|---|
| Metrics | PSNR | SSIM | PSNR | SSIM | PSNR | SSIM | PSNR | SSIM | PSNR | SSIM |
| DSC | 27.16 | 0.8663 | 14.73 | 0.3815 | 24.24 | 0.8279 | 27.31 | 0.8373 | 34.95 | 0.9416 |
| GMM | 28.66 | 0.8652 | 14.50 | 0.4164 | 25.81 | 0.8344 | 27.55 | 0.8479 | 34.30 | 0.9428 |
| DDN | 34.68 | 0.9671 | 26.05 | 0.8056 | 30.97 | 0.9116 | 30.00 | 0.9041 | 36.16 | 0.9457 |
| RESCAN | 36.09 | 0.9697 | 26.75 | 0.8353 | 33.38 | 0.9417 | 31.94 | 0.9345 | 38.11 | 0.9707 |
| PReNet | 37.80 | 0.9814 | 29.04 | 0.8991 | 33.17 | 0.9481 | 32.60 | 0.9459 | 40.16 | 0.9816 |
| MSPFN | 38.58 | 0.9827 | 29.36 | 0.9034 | 33.72 | 0.9550 | 32.99 | 0.9333 | 43.43 | 0.9843 |
| RCDNet | 39.17 | 0.9885 | 30.24 | 0.9048 | 34.08 | 0.9532 | 33.04 | 0.9472 | 43.36 | 0.9831 |
| MPRNet | 39.47 | 0.9825 | 30.67 | 0.9110 | 33.99 | 0.9590 | 33.10 | 0.9347 | 43.64 | 0.9844 |
| DualGCN | 40.73 | 0.9886 | 31.15 | 0.9125 | 34.37 | 0.9620 | 33.01 | 0.9489 | 44.18 | 0.9902 |
| SPDNet | 40.50 | 0.9875 | 31.28 | 0.9207 | 34.57 | 0.9560 | 33.15 | 0.9457 | 43.20 | 0.9871 |
| Uformer | 40.20 | 0.9860 | 30.80 | 0.9105 | 35.02 | 0.9621 | 33.95 | 0.9545 | 46.13 | 0.9913 |
| Restormer | 40.99 | 0.9890 | 32.00 | 0.9329 | 35.29 | 0.9641 | 34.20 | 0.9571 | 47.98 | 0.9921 |
| IDT | 40.74 | 0.9884 | 32.10 | 0.9344 | 34.89 | 0.9623 | 33.84 | 0.9549 | 47.35 | 0.9930 |
| DRSformer | 41.23 | 0.9894 | 32.17 | 0.9326 | 35.35 | 0.9646 | 34.35 | 0.9588 | 48.54 | 0.9924 |
| **Regformer (ours)** | 41.51 | 0.9900 | 32.46 | 0.9353 | 35.43 | 0.9651 | 34.38 | 0.9591 | 48.60 | 0.9941 |

initial channel to 48 for Regformer. During training, we employ the AdamW Loshchilov & Hutter (2019) optimizer with parameters $\beta1 = 0.9$, $\beta2 = 0.999$, and weight decay = 1e-4. In the experiment part, we set $n = 2$, $k_1 = 3$ and $k_2 = 5$ in the MGFB. Please note that in different tasks or different datasets, we may modify these hyperparameters to achieve better results. Concurrently, we follow DRSformer's settings Chen et al. (2023b), which utilize the L1 loss as the loss function. We set the initial learning rate to 3e-4 and apply a cosine annealing algorithm to gradually decay it to 1e-6. The patch size is set to $128 \times 128$, and we perform a total of 300K iterations during training using two NVIDIA GeForce RTX 3090 GPUs with 24GB memory each. Any settings that are not explicitly mentioned in this paper keep consistent with those used in DRSformer Chen et al. (2023b).

## 4.2 DERAINING PERFORMANCE EVALUATIONS

**(1) Quantitative Results.** The comparison results of our model and other related methods on different datasets are shown in Table 1 and Table 2. Specifically, in Table 1, we show the results on rain streak datasets, while Table 2 shows our results on the raindrop dataset. For **rain streak removal**, we can clearly see that the PSNR and SSIM values of our method on each dataset exceed the existing solutions, or maintain a high competitive level. Especially on the Rain200L and Rain200H datasets, our method has greatly improved the restored results, and outperformed current methods by up to 0.26 dB and 0.28 dB. On other datasets, PSNR far exceeds the baseline Zamir et al. (2022), and at the same time refreshes the current SOTA level. This clearly demonstrates the effectiveness of our innovative approach to Regformer. In addition, it

| Metrics | PSNR | SSIM |
|---|---|---|
| Eigen *et al.* Eigen et al. (2013) | 21.31 | 0.7572 |
| Pix2Pix Isola et al. (2017) | 27.20 | 0.8359 |
| AttentGAN Qian et al. (2018) | 31.59 | 0.9170 |
| Quan *et al.* Quan et al. (2019) | 31.37 | 0.9183 |
| IDT Xiao et al. (2022) | 31.87 | 0.9313 |
| **Regformer (ours)** | 32.51 | 0.9725 |

Table 2: Quantitative comparison with other related methods for single image raindrop removal on AGAN-Data Qian et al. (2018). The best and second best results are colored with red and blue.

can be seen from the chart that the CNN-based method still maintains high competitiveness compared with the current Transformer-based method, but the performance of methods based on prior knowledge, such as DSC Luo et al. (2015) and GMM Li et al. (2016), is relatively backward. Although DRSformer also performs effective attention mechanism exploration based on Restormer, they lag behind our Regformer on both synthetic and real datasets since they do not explore the difference between different regions. In addition, for **raindrop removal**, we can clearly see that our method far exceeds current related methods by up to 0.64 dB, and the error rate decreased by nearly 60% (the ground truth's SSIM is 1.0000).

**(2) Visualization Results.** We present our rain streak removal results in detail in Figures 2 and 5. Notably, we present a visualization of our model's ideas in Figure 1, 6 and 7, which focus on showing why and how our model works. Figures 2, 5 show the comparison results of ours and other existing representative methods on the real dataset SPA-Data, respectively. Clearly, we can see that in some areas, other compared methods do not remove the rain streak information well, or produce image distortion problems while removing the rain streak information. Different from other methods, our method preserves the original details of the image content area while removing the rain streaks, which fully demonstrates the superiority of our method. Due to space limitations, more visualization results will be shown in the appendix section.

### 4.3 ABLATION STUDIES

In the ablation study, we utilize the Rain200L dataset to train and evaluate our network. We take Restormer Zamir et al. (2022) as the Baseline and modify some of its parameters to enhance its performance. In Table 3, Baseline$_1$ (v1) denotes the initial Restormer; baseline$_2$ (v2) denotes the Restormer with optimized parameters; w/RTC$_1$ (v3) refers to the addition of RTC without employing the RegionMask mechanism (i.e., no mask operation); and w/RTC$_2$ (v4) signifies the incorporation of the RegionMask mechanism. The term w/MGFB (v5) represents the integration of the MGFB. This ablation study aims to provide a clear understanding of the individual component's contributions and the overall effectiveness of our proposed Regformer model.

**(1) Effectiveness of Training Settings.** Initially, we train our model using the default settings of Restormer (v1) and subsequently retrain it by adjusting some of its parameters (v2). The results can be seen in Table 3. For specific parameter details, please refer to the training details section. We fine-tune the number of layers and attention heads within each module to better adapt our model to the target task and establish a solid foundation for further modifications. The modifications in training settings serve to enhance our model's performance, making it more effective in handling SID task.

**(2) Effectiveness of RTC and Region Mask Mechanism.** Table 3 presents the results of incorporating the RTC and region mask mechanisms into our model. Clearly, **adding the RTC module without the region mask mechanism** does not significantly improve the model's performance

|    | Variants | PSNR | SSIM |
|----|----------|------|------|
| v1 | Restormer $_1$ | 40.99 | 0.9890 |
| v2 | Baseline $_2$ | 41.24 | 0.9897 |
| v3 | w/RTC$_1$ | 41.25 | 0.9897 |
| v4 | w/RTC$_2$ | 41.42 | 0.9899 |
| v5 | w/MGFB | 41.26 | 0.9898 |
| v6 | v5+FGMask | 41.46 | 0.9899 |
| v7 | v5+BGMask | 41.43 | 0.9899 |
| v8 | **Regformer** | **41.51** | **0.9900** |

Table 3: Ablation study on Rain200L dataset. It reflects the role of each component in our Regformer.

(compared to v2, v3 exhibits only a 0.01 dB increase). We add foreground mask (v6) and background mask (v7), respectively, and the performance improves by 0.20dB and 0.17dB, respectively. However, upon **integrating the entire Region Mask mechanism**, the network effectively captures feature information from different regions. This allows the model to efficiently comprehend the features of distinct regions and their fused attributes, thereby achieving a substantial performance enhancement (compared to v3, the performance of v4 improves by 0.17 dB; and relative to v5, the performance of Regformer is boosted by 0.25 dB). These results highlight the crucial role the region mask mechanism plays in Regformer, demonstrating its effectiveness in handling single image deraining task.

**(3) Effectiveness of MGFB.** Table 3 illustrates the impact of incorporating MGFB into our model. As evident from the table, adding MGFB without introducing RTC (v5) does not significantly enhance the model's restoration performance (compared to v2, the PSNR of v5 increases by a mere 0.02 dB). This outcome can be attributed to MGFB being primarily designed as a mixed-scale feature extraction mechanism for different Region Mask attention mechanisms, intended to thoroughly extract features across various scales and receptive fields in distinct regions. Without employing the Region Mask attention mechanism, MGFB is restricted to extracting information from a single-layer region, which limits its performance. However, when RTC$_2$ is integrated, we see a notable performance improvement (Regformer's PSNR is enhanced by 0.09 dB compared to v4), which directly substantiates the validity and rationality of our approach. This highlights the effectiveness of MGFB within our proposed framework for SID.

### 5 CONCLUSION

In this paper, we introduced Regformer, a novel SID model that combines the advantages of both the rain-affected and rain-unaffected regions. The Regformer model efficiently integrates the local and non-local features to effectively reconstruct high-quality images while preserving essential details in both the rain-affected and unaffected regions. Our proposed embedded region mask mechanism allows the model to selectively focus on crucial information from both regions. By employing rain masks and unaffected region masks, the model can better differentiate between the two regions and ultimately yield a feature map with region-specific features. Extensive experiments on rain streak and raindrop removal demonstrated the effectiveness of our model. The results showed that our model outperforms current methods in terms of both quantitative and qualitative evaluations. In conclusion, Regformer is a promising SID model, achieving superior performance by leveraging the power of Transformer and U-Net architectures while incorporating an effective region mask mechanism. Future work may involve exploring additional masking strategies for further improvement and extending the model to other image restoration tasks, including dehazing and denoising.

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

## 6 APPENDIX

In this section, we mainly show the visual results of using the Mask mechanism to process images, and more visual results of image deraining on Rain200H Yang et al. (2017) and SPA-Data Wang et al. (2019) datasets.

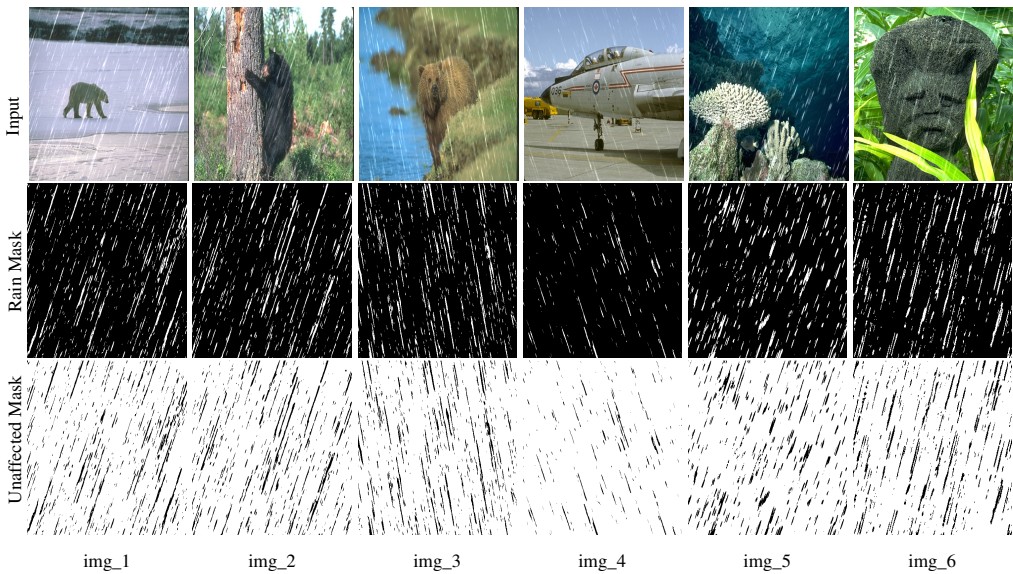

Figure 6: The visualization image results of the rain mask and unaffected region mask we generated on Rain200L dataset Yang et al. (2017). As can be seen in the figure, most of the rain region and the unaffected region has been successfully distinguished.

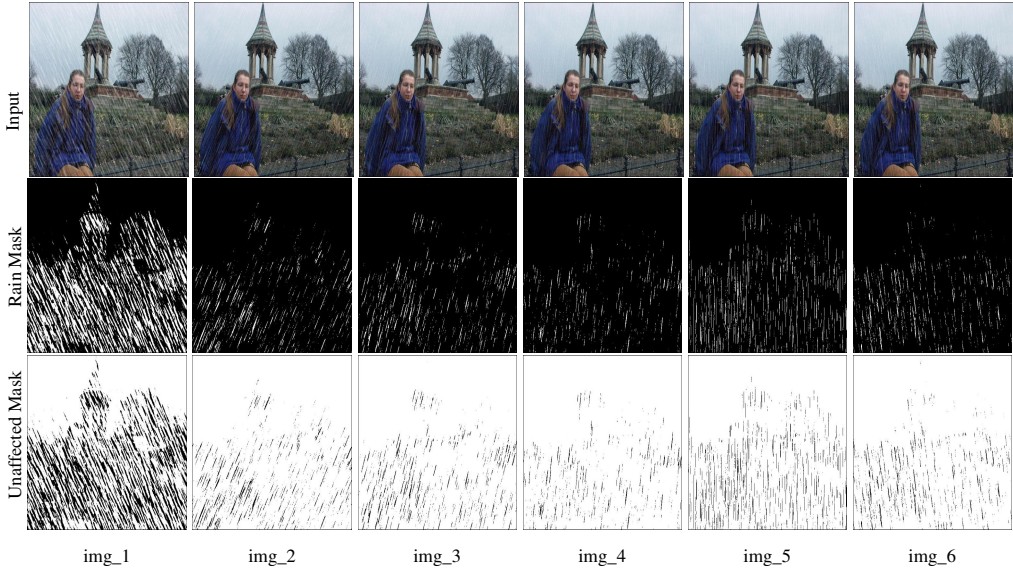

Figure 7: The visualization image results of the rain mask and unaffected region mask we generated on DDN dataset Fu et al. (2017). They are one image with rain content added in different directions and different levels. As can be seen in the figure, most of the rain region and the unaffected region has been successfully distinguished.

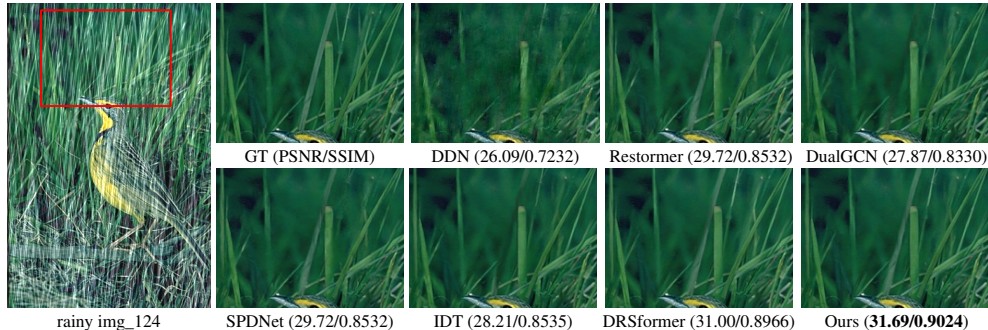

Figure 8: Visual comparison of each method on Rain200H Yang et al. (2017) dataset. Clearly, our Regformer model can perform more accurate detail and texture recovery, compared to other methods.

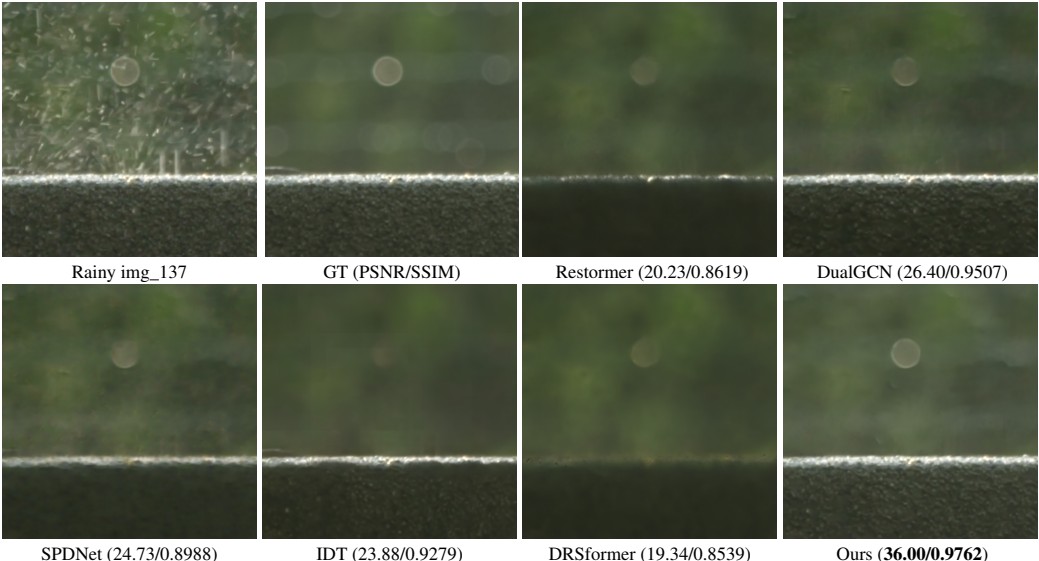

Figure 9: Visual comparison of each method on SPA-Data Wang et al. (2019) dataset. As can be seen in the figure, our model preserves more precise details while removing rain streak.

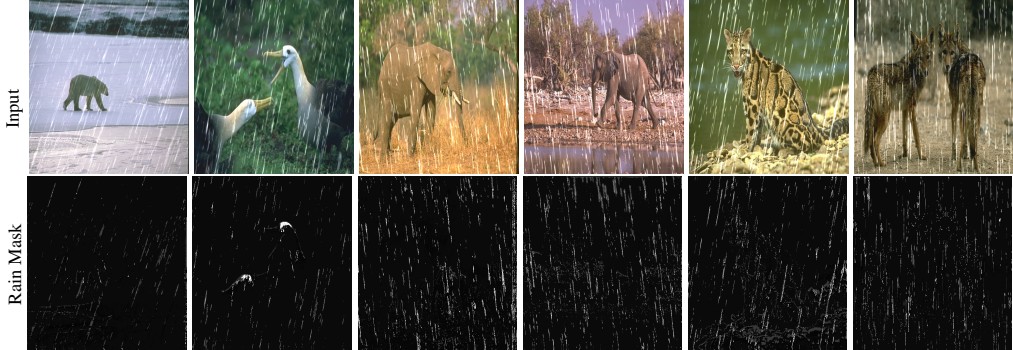

Figure 10: These mask feature maps on Rain200L dataset Yang et al. (2017) represent the masks as they are processed within the network during inference. To effectively illustrate our high-quality mask, we have averaged the masks across their respective channels and transformed them into RGB images for a more direct and visual representation.

