# OpenReview forum: "Diving Deep into Regions: Exploiting Regional information Transformer for Single Image Deraining"
_ICLR.cc/2024/Conference — Submitted to ICLR 2024_

### Official Review · Reviewer_29g8 · 2023-10-29

**Soundness:** 3 good
**Presentation:** 4 excellent
**Contribution:** 3 good
**Rating:** 8
**Confidence:** 5

**Summary:**

This paper proposes a region-aware Transformer for single image deraining, which consists of a region masked attention and a mixed gate forward block. The authors observe that existing deraining methods ignore the differences between rain-affected and unaffected regions.

**Strengths:**

S1. The design of region masked attention is simple and effective, which can process rain-affected and unaffected regions of images separately.

S2. The experimental evaluation and discussion are adequate, and the results convincingly support the main claims.

S3. The paper is well-organized and clearly written.

**Weaknesses:**

W1. The author should delve into the importance of the rain-affected and unaffected regions of rainy images. I am curious if utilizing the features of he unaffected regions can better guide image restoration?

W2. How to determine whether high-quality region masks can be generated? If some visual examples of intermediate feature maps that illustrate this property are added, the proposed method can become compelling.

-----------------------After Rebuttal---------------------------

Thank you for your feedback. The rebuttal addressed my concerns well. Considering other reviews, I decide to keep my score as Accept.

**Questions:**

See the above Weaknesses part.

---

> ### Author Response · Authors · 2023-11-18
> **Response to Reviewer 29g8, and thanks for your comments.**
>
> Thank you for reviewing our paper.
>
> ---
> ***Q1:*** The author should delve into the importance of the rain-affected and unaffected regions of rainy images. I am curious if utilizing the features of the unaffected regions can better guide image restoration?
>
> ***Response:*** Thanks for your concerns. First, let me explain why we use rain-affected(Foreground) regions:
>
> **a)Foreground Mask (rain affected mask):** We utilize the foreground mask to better extract the characteristics of foreground rain streaks. This enables the network to identify the varying rain patterns more effectively, preventing the erroneous removal of rain-like features (as shown in Figure 2, where light spots are mistakenly removed as the rain for the results of DRSformer, IDT, and DualGCN).
>
> **b) Background Mask (rain unaffected mask):** The employment of the background mask aims to emphasize useful features, which can be utilized to recover details in rain streaks (as depicted in Figure 7 in the appendix, where rain streaks in the sky are not separated for the ablation setting without the background mask). This ensures that rain areas can be effectively recovered with desired content via the following ``combing mask" stage.
>
> **c) Combining Masks:**
> As shown in Figure 1 of the main paper, a full mask is utilized to incorporate the results from both masks with a transformer block. This technique allows the non-rain areas in the background to guide the restoration of the corresponding rain-streaked foreground areas, resulting in more realistic image details.
>
>
> We have conducted experiments using foreground and background masks, respectively, in the ablation experiment section. The specific details are explained in the ablation study section (Sec. 4.3).
> The results, particularly for variants v6, v7, and v8, demonstrate the efficacy of our strategy.
> In addition, we also conducted partial ablation experiments on the SPA-Data dataset, as shown below. The results again validate our approach.
>
> ---
> |    | Variants         | PSNR  | SSIM   |
> |----|------------------|-------|--------|
> | v2 | Baseline $_{2}$  | 44.38 | 0.988 |
> | v4 | w/RTC$_{2}$      | 44.84 | 0.990 |
> | v5 | w/MGFB           | 44.89 | 0.988 |
> | v6 | v5+FGMask        | 45.25 | 0,990 |
> | v7 | v5+BGMask        | 45.11 | 0.989 |
> | v8 | **Regformer**    | 45.41 | 0.990 |
>
> ---
>
> Besides, **Reviewer ac1q** also agrees with the significance of using foreground mask and background separately. He mentioned that our method first points out the importance of explicitly and individually processing rain-affected and unaffected regions and then combing them.
>
>
> Finally, we appreciate your comments and welcome further questions or clarifications. We are committed to ongoing discussions to enhance the quality of our paper and address any concerns you may have.
>
>
> ---
>
> ***Q2:*** How to determine whether high-quality region masks can be generated? If some visual examples of intermediate feature maps that illustrate this property are added, the proposed method can become compelling.
>
> ***Response:***
>
> Thank you for your valuable suggestion. In response to your query on the quality of generated region masks, we have included a series of intermediate feature maps in the revised supplementary materials, specifically showcased in Figure 10. These feature maps represent the masks as they are processed within the network during inference. To effectively visualize them, we have averaged the masks across their respective channels and transformed them into RGB images for a more direct and visual representation.
>
> We believe this addition will provide a clearer insight into the quality and functionality of the masks generated by our method, thereby bolstering the credibility of our proposed approach.
>
> Should you have any further questions or additional suggestions, we are more than open to them. Your feedback is instrumental in enhancing the quality of our work, and we greatly appreciate your contribution to this process.

---

### Official Review · Reviewer_ac1q · 2023-10-30

**Soundness:** 4 excellent
**Presentation:** 3 good
**Contribution:** 4 excellent
**Rating:** 8
**Confidence:** 5

**Summary:**

This paper proposes a novel and effective transformer-based method for Single Image Deraining (SID). In this paper, one significant motivation is the consideration of processing rain-affected and unaffected regions independently and then combining their effects for the final reconstruction. Such a hierarchical strategy helps to vastly improve the deraining effects, especially in these challenging cases. The proposed framework is called Region Transformer (Regformer), consisting of Region Masked Attention (RMA) mechanism and a Mixed Gate Forward Block (MGFB). Extensive experiments are conducted on all current representative deraining datasets, and both RMA and MGFB are verified to be effective in dealing with deraining degradations.

**Strengths:**

1.	The proposed framework first points out the importance of explicitly and individually processing rain-affected and unaffected regions, leading to new SOTA performance on all current SID datasets, without the increase of model parameters/flops compared with existing methods.
2.	Compared with current baselines, the visual results of Regformer are much better, especially in these regions with details while covered with the raindrop in the original image. Its performance for different real-world cases has also proven to be great.
3.	The writing and organization of this paper is satisfactory.

**Weaknesses:**

1.	More perceptual metrics can be added for comparison in Table 1 with strong baselines, like LPIPS, which can reflect the quality of the restoration from different aspects.
2.	The ablation study in Table 3 can be conducted on more datasets to get a more comprehensive analysis.

**Questions:**

1.	What is the effect of $k_i$ in MGFB?
2.	What are the implement details of baselines? i.e., how their scores are obtained in Table 1? These should be described in detail.

---

> ### Author Response · Authors · 2023-11-18
> **Response to Reviewer ac1q, and thanks for your comments. (PART I)**
>
> Thank you for reviewing our paper.
>
> ---
>
> ***Q1:*** More perceptual metrics can be added for comparison in Table 1 with strong baselines, like LPIPS, which can reflect the quality of the restoration from different aspects.
>
> ***Response:*** Thank you for your valuable suggestion to include more perceptual metrics in our comparative analysis. We recognize the importance of such metrics, like LPIPS (Learned Perceptual Image Patch Similarity), in providing a more holistic view of image restoration quality.
>
> In response to your recommendation, we have evaluated the performance of our method, Regformer, alongside other established methods using these metrics on the SPA-Data dataset. The results are summarized in the table below:
>
> ---
> | Model         | LPIPS  |
> |------------------|-------|
> | Restormer | 0.0278|
> |  DRSformer    | 0.0267 |
> | Regformer          | **0.0247** |
>
>
> ---
>
> We will ensure these additional metrics are included in supplementary materials of our paper.
>
> These metrics should provide a more nuanced understanding of each method's performance in terms of perceptual image quality.
>
> ---
>
> ***Q2:*** The ablation study in Table 3 can be conducted on more datasets to get a more comprehensive analysis.
>
> ***Response:***
> Thank you for your suggestion regarding the expansion of our ablation studies to include more datasets. In response to your comments and the feedback from reviewer FN5q, we have extended our ablation experiments to the SPA-Data dataset. This additional analysis aligns with our goal to provide a more comprehensive evaluation of our model.
>
> The table below presents the results of these extended ablation studies on the SPA-Data dataset. For a swifter qualitative assessment, these models were retrained with 100k iterations, though it is important to note that this performance is not as robust as the 300k iterations results detailed in our paper. We selected several major ablation experiments for training and testing to highlight critical aspects of our method:
>
>
> |    | Variants         | PSNR  | SSIM   |
> |----|------------------|-------|--------|
> | v2 | Baseline $_{2}$  | 44.38 | 0.988 |
> | v4 | w/RTC$_{2}$      | 44.84 | 0.990 |
> | v5 | w/MGFB           | 44.89 | 0.988 |
> | v6 | v5+FGMask        | 45.25 | 0,990 |
> | v7 | v5+BGMask        | 45.11 | 0.989 |
> | v8 | **Regformer**    | 45.41 | 0.990 |
>
> ---
>
>
> The advantage of our full setting (**Regformer**) compared with these ablation settings again validates the effectiveness of each component in our approach.
>
> Should you have any further questions or need additional clarification, please let us know. We appreciate your feedback and are committed to ensuring the thoroughness and accuracy of our research.

---

> ### Author Response · Authors · 2023-11-18
> **Response to Reviewer ac1q, and thanks for your comments. (PART II)**
>
> ***Q3:*** What's the effect of $k_{i}$ in MGFB?
>
> ***Response:*** Thank you for your question regarding the role of $k_i$ in the MGFB module. Here $n$ and $k_i$ are hyper-parameters that will influence the performance of MGFB.
>
> We conducted further experiments with varying $n$ and $k_i$, along with different activation function placements on the Rain200L Dataset. For a quicker quantitative assessment, we retrained these models with 100k iterations. Thus, it is noteworthy that this performance is lower compared to the 300k iteration results presented in the paper. The results are as follows:
>
> ---
> |    | Variants                       | PSNR  | SSIM   |
> |----|------------------|-------|--------|
> | vv1 | $N=2,k_1=3,k_2=3,$ activate $k_1$         | 40.42 | 0.9879 |
> | **Regformer**|$N=2,k_1=3,k_2=5,$ activate $k_1$          | **40.52** | **0.9880** |
> | vv3 | $N=3, k_1=3, k_2=5, k_3=7, $ activate $k_1$| 40.44 | 0.9877   |
> | vv4 | $N=3, k_1=3, k_2=5, k_3=7, $ activate $k_1, k_2$| 40.42 | 0.9876 |
> ---
>
> These results demonstrate the impact of varying $k_i$ values and activation function placements on the model's performance.
>
>
> ***Q4:*** What are the implement details of baselines? i.e., how their scores are obtained in Table 1? These should be described in detail.
>
> ***Response:*** Thank you for raising the important question regarding the implementation details of the baseline models used in our study, particularly in relation to the scores presented in Table 1.
>
> For a fair comparison, we following the experimental setting in the DRSformer[1] paper.
> We ensured that the experimental environment and other variables were kept consistent to maintain the integrity of the comparison. This approach allowed us to directly use some baseline scores as reported in the DRSformer paper.
>
> Furthermore, for these methods without scores to refer to, we re-train them.
> For the sake of fairness and accuracy, all experimental settings, except for the specific details pertaining to our model parameters, were aligned with those used in the DRSformer study. This consistency is critical to ensure that any observed differences in performance are attributable to the models themselves rather than variations in the experimental setup.
>
> We appreciate your suggestion and will make sure to further clarify and emphasize this aspect in the experimental section of our paper. This will help ensure transparency and allow readers to understand the context under which the baseline comparisons were made.
>
> If you have any further questions or need additional information, please feel free to reach out. We are committed to maintaining rigorous standards in our research methodology and welcome any feedback that can help enhance the clarity and quality of our work.
>
> [1] DRSformer: Learning a Sparse Transformer Network for Effective Image Deraining. CVPR 2023.
>
> ---

---

### Official Review · Reviewer_FN5q · 2023-10-30

**Soundness:** 2 fair
**Presentation:** 2 fair
**Contribution:** 2 fair
**Rating:** 6
**Confidence:** 5

**Summary:**

This paper dives into single image deraining in the aspect of rain-affected regions and unaffected regions, and tries to remove rain streaks and preserve background parts. Based on this motivation, it proposes Region Transformer Block, which is composed of a Region Masked Attention mechanism and a Mixed Gate Forward Block. The former takes the information of rain-affected regions and unaffected regions into consideration and generates attention maps with region masks. The latter utilizes different kernel sizes to extract features on different receptive fields. It achieves SOTA results on multiple datasets.

**Strengths:**

1. More explicit decomposition of rain removal issues into rain streak removal in rain-affected regions and detail  preservation in unaffected regions
2. Better performance with fewer parameters and equivalent computation cost compared with Restormer and DRSformer.

**Weaknesses:**

1. The main concern is that this approach may be only effective on synthetic datasets. As shown in Table 1, RegFormer only brings 0.06dB PSNR gain on real-world dataset SPA-Data compared to DRSFormer. And when testing on a more realistic dataset (such as WeatherStream [1]), I'm worried that this approach may offer little improvement.
2. The generation of region mask need to be further clarified. The expression in Sec. 3.2.1 seems to conflict with Figure 3.
3. Could you provide the ablation studies (similar to Table 3) on SPA-Data dataset?
4. It would be better if comparisons of inference time were given.

[1] WeatherStream: Light Transport Automation of Single Image Deweathering. CVPR 2023.

**Questions:**

Please see 'Weaknesses'.

---

> ### Author Response · Authors · 2023-11-18
> **Response to Reviewer FN5q, and thanks for your comments. (PART I)**
>
> Thank you for reviewing our paper.
>
> ---
>
> ***Q1:*** The main concern is that this approach may be only effective on synthetic datasets. As shown in Table 1, RegFormer only brings 0.06dB PSNR gain on real-world dataset SPA-Data compared to DRSFormer. And when testing on a more realistic dataset (such as WeatherStream [1]), I'm worried that this approach may offer little improvement.
>
> ***Response:***
>
> Thank you for expressing your concerns regarding the effectiveness of our approach on real-world datasets.
> To address this, I'd like to offer additional insights:
>
> First, the quantitative metric of PSNR is global and, as such, may not effectively reflect local variations in image content. Our paper, illustrated in Figures 2, 5, and 9, demonstrates that our results surpass the baselines on real-world SPA-data dataset, in terms of human perceptions. The visual representations in these figures provide a clearer appreciation of the enhancements offered by our method.
>
> Second, in response to your suggestion, we tested our method on the WeatherStream Dataset. For quicker quantitative assessment, these models were retrained for 100k iterations. Please note this performance is lower than what would be observed with the standard training duration. Here are the comparative results with two prominent transformer-based models, Restormer[1] and DRSformer[2]. Note that for quicker evaluation, we use datasets in train2.zip to train these three models.
>
> ---
>
> | Methods         | PSNR  | SSIM   |
> |------------------|-------|--------|
> | Restormer | 24.02 | 0.7501 |
> | DRSformer | 24.13 | 0.7521 |
> |  Regformer (ours) | 24.16 | 0.7532 |
>
> ---
>
> It is evident that our performance remains superior to these strong baselines on the WeatherStream dataset. This superiority aligns with the efficacy of the mask mechanism incorporated in our approach. It allows for more accurate processing of non-rainy regions during rain isolation, significantly boosting the discriminative capability of our model in handling rain-like image content features. This aspect has also been noted by reviewer ac1q.
>
> We plan to include these findings in an updated version of the supplementary materials and will cite the WeatherStream paper accordingly.
>
> ---
>
> ***Q2:*** The generation of region mask need to be further clarified. The expression in Sec. 3.2.1 seems to conflict with Figure 3.
>
> ***Response:*** Thank you for highlighting the inconsistency between Section 3.2.1 and Figure 3 in our paper, specifically regarding the generation of the region mask. Your observation is invaluable, and we appreciate your meticulous attention to detail.
>
> Upon a detailed review, we realized an oversight in Figure 3(a). The depiction suggested that the input image, after undergoing a 3x3 convolution, would yield three types of masks, which could potentially lead to confusion about where exactly the mask is generated. We have now corrected Figure 3 in the revised paper.
>
> These features after 3x3 convolution will be employed in the decoder's RTC, and the difference between these shallow features and the decoder's restored features at varying levels will lead to mask computation.
> As elaborated in Figure 4 of the main paper, the mask is computed by conducting threshold on the feature map.
> For ease of understanding, we show images mapped to RGB space rather than feature maps.
>
> ---
>
> [1] WeatherStream: Light Transport Automation of Single Image Deweathering. CVPR 2023.

---

> ### Author Response · Authors · 2023-11-18
> **Response to Reviewer FN5q, and thanks for your comments. (PART II)**
>
> ***Q3:*** Could you provide the ablation studies (similar to Table 3) on SPA-Data dataset?
>
> ***Response:*** Thank you for your request to provide ablation studies on the SPA-Data dataset, similar to those presented in Table 3. We value your suggestion and have conducted several key ablation experiments to fulfill your request. Please note that due to constraints in time and resources, we have focused on testing some ablation settings with higher performance in Table 3.
>
> For a quicker qualitative assessment, we have retrained these models for 100k iterations. It's important to mention that this performance metric is lower than the full 300k iterations results detailed in our paper. The following table illustrates the outcomes of our ablation studies on the SPA-Data Dataset:
>
> |    | Variants         | PSNR  | SSIM   |
> |----|------------------|-------|--------|
> | v2 | Baseline $_{2}$  | 44.38 | 0.988 |
> | v4 | w/RTC$_{2}$      | 44.84 | 0.990 |
> | v5 | w/MGFB           | 44.89 | 0.988 |
> | v6 | v5+FGMask        | 45.25 | 0,990 |
> | v7 | v5+BGMask        | 45.11 | 0.989 |
> | v8 | **Regformer**    | 45.41 | 0.990 |
>
> ---
>
> Our full setting (**Regformer**) continues to exhibit superior performance compared to these ablation settings, consistent with the findings presented in Table 3 of the main paper.
> We have selected these particular variants for their relevance and impact in demonstrating the effectiveness of our proposed method.
> These results will be provided in the final version of our paper.
>
>
> ---
>
> ***Q4:*** It would be better if comparisons of inference time were given.
>
> ***Response:*** Thank you for your valuable suggestion regarding the comparison of inference times. We have conducted a detailed analysis to compare the inference times of our model, RegFormer, with two strong models we have compared against in our paper, including Restormer [1] and DRSformer [2].
>
>
> To ensure the comparison accuracy, we compute the average inference time with 300 images, each with the shape of [1, 3, 256, 256].
> The following table presents the inference time (in milliseconds) for each model:
>
> -----
>
> | Model         | Inference Time(ms)  |
> |------------------|-------|
> |  Restormer | 80.0 |
> | DRSformer | 168.7 |
> | Regformer(ours)   | 138.8 |
>
> ------
>
> These results demonstrate the efficiency of our Regformer model in comparison to the existing models. It's important to note that while our model is faster than DRSformer, it is slightly slower than Restormer. We believe this is a reasonable trade-off considering the performance gains our model offers.
>
> We will include these details in the final version of our paper. If you have any further questions or need more information, please let us know. We are open to further discussions and appreciate your feedback in enhancing the quality of our research.
>
> [1] Restormer: Efficient Transformer for High-Resolution Image Restoration. CVPR 2022.
>
> [2] DRSformer: Learning a Sparse Transformer Network for Effective Image Deraining. CVPR 2023.

---

> ### Author Response · Authors · 2023-11-21
> **Waiting for your further feedbacks.**
>
> Dear Reviewer FN5q:
>
> We have addressed the questions and concerns raised during the review process and submitted our responses. We would like to know if you have further questions or additional feedback on our revisions. Your comments are invaluable to us.
>
> Best regards,
>
> Authors of paper 1110.

---

> > ### Comment · Reviewer_FN5q · 2023-11-22
> > **Thanks**
> >
> > Many thanks. I still have concerns that this approach may be only effective on synthetic datasets, as the experiments on WeatherStream show it can only achieve little PSNR gain. Thus, I keep my rating.

---

> > > ### Author Response · Authors · 2023-11-23
> > > **Response to Reviewer FN5q, and thanks for your comments.**
> > >
> > > Thank you for your feedback. We understand your concerns regarding the effectiveness of our approach, particularly highlighted by the modest PSNR gains on the WeatherStream dataset. Due to the impending rebuttal deadline, we were unable to conduct experiments on a broader range of datasets right now. However, we encourage a more holistic evaluation of our model's performance beyond just the PSNR metric.
> > >
> > > While it is true that our model shows a limited PSNR improvement of 0.03dB over DRSformer, it is crucial to consider the significant reduction in model complexity. Our model requires only 22M parameters compared to DRSformer's 33M, a reduction of approximately 33\%. Additionally, our approach utilizes fewer FLOPs, as illustrated in Figure 1(c).
> > >
> > >
> > > Our approach not only achieves performance improvements over DRSformer but also does so with significantly fewer parameters. This efficiency is a critical aspect of our contribution. We posit that aligning our model's parameter count with that of DRSformer, for instance, by increasing the number of blocks, could potentially lead to even more significant performance gains.
> > >
> > >
> > > Furthermore, on the additional RainDrop dataset (a real-world dataset), we observed a notable improvement of 0.64dB in PSNR, 0.04 in SSIM over IDT, as shown in Table 2.
> > >
> > > We believe these enhancements are substantial and demonstrate the efficacy of our model in more practical scenarios.
> > >
> > >
> > > We appreciate your insights and remain open to further discussion to improve our work. Thank you for your
> > > consideration.

---

> > > > ### Comment · Reviewer_FN5q · 2023-11-23
> > > > **Thanks**
> > > >
> > > > Many Thanks. I think this method may be effective for removing raindrops and rain streaks, while less effective for removing rain mist. I will carefully make the final decision based on the opinions of other reviewers.

---

### Official Review · Reviewer_edgT · 2023-11-01

**Soundness:** 3 good
**Presentation:** 3 good
**Contribution:** 3 good
**Rating:** 5
**Confidence:** 5

**Summary:**

This paper introduces a regional basded transformer network to tackle single image deraining problem. The proposed method includes a new architecture called Region Transformer Block, which utilizes the power of a masked attention structure and a mixed gated forward component. The region transformer is trying to learn features from the non-rain region to better recover the rain affected parts. Extensive experiments show that the proposed method outperforms the baseline methods in a consistent manner.

**Strengths:**

1. This paper is well written with strong motivation of solving the deraining problem by learning the unique features from the same image.
2. The proposed method consistently outperforms others on benchmarking datasets.

**Weaknesses:**

1. In Eq. (6), the big \Pi is  indicating that the result of element-wise addition are multiplied together to form feature F. It is not stated clearly  that how the multiplication should be done. And why using multiplication?
2. The last line on page 6, what does it mean by n and k_i are all parameters?

**Questions:**

One of the most important question in this paper is regarding the novelty. The region-based attention mechanism has been applied by many previous works in various areas. The proposed method does not show the advanced benefits of using the masked attention on the deraining problem, especially the explicit mechanism / design to identify the true features of non-rain regions.

The authors are suggested to answer the question and the weaknesses during the rebuttal period.

---

> ### Author Response · Authors · 2023-11-15
> **Response to Reviewer edgT, and thanks for your comments. （PART I)**
>
> Thank you for taking the time to review our paper.
>
> ***Q1:*** In Eq. (6), the big \Pi is indicating that the result of element-wise addition are multiplied together to form feature $F$. It is not stated clearly that how the multiplication should be done. And why using multiplication?
>
> ***Response:*** Thank you for your insightful comments. Based on your description, I believe you are referring to Eq. (5) in our paper. Upon re-examination, prompted by your remarks, we identified a necessary correction in our manuscript regarding this equation.
>
>  **(1) Correction in Equation Description:**
>  In our original manuscript, Equation (5) was described as:
>
>  $ F = \prod_{i=1}^{n}(DWConv_{k_{i} \times k_{i}}(M) + M)$.
>
> However, we have amended this as follows to enhance clarity and accuracy:
>
>  $F = Activation(DWConv_{k_{1}\times k_{1}}(M) + M) \odot \prod_{i= 2}^{n}(DWConv_{k_{i}\times k_{i}}(M) + M) $.
>
> Concurrently, we have removed the activation function from Equation (6). In summary, this modification shifts the activation function's position to precede the matrix multiplication. We have updated the correction in our newly revised paper (in Section 3.2.2, we use red highlighted text to display). This revision, sparked by your observation, led us to re-examine our code, ensuring the paper's precision. We aim to diligently review the manuscript before final publication to preclude similar issues.
>
> **(2) Rationale Behind Multiplication Usage:**
> This adjustment signifies that we conduct element-wise multiplication across the outcomes of all parallel branches, as detailed in the revised Equation (5). This strategy is akin to the gating mechanism in Restormer[1], where feature maps from two parallel branches are multiplied to specialize each branch's feature processing ability before fusion. Equation (5) is the extension of such a gating strategy for multiple branches, enabling each to focus on a unique feature scale, propagating the detailed aspects complementary to other branches. We have updated this description using the red font in Section 3.2.2.
>
> We appreciate your valuable feedback and guarantee to incorporate these changes in the paper's final version.
>
> ---
>
> ***Q2:***  The last line on page 6, what does it mean by n and k_i are all parameters?
>
> ***Response:*** Thank you for your kind question. Let me clarify the statement about $n$ and $k_i$ being parameters.
> In our research, $n$ and $k_i$ are indeed hyperparameters that we have the discretion to select manually. Our intent in designing the model in this manner was to transcend the conventional gating solutions typically seen in works like Restormer[1] and DRSformer[2]. By adjusting the FFN structure, we aimed to offer a more versatile and adaptable model with multiple scales. This design provides users with a broader range of suitable options when dealing with varied datasets or distinct tasks. We have updated this description using red font in section 4.1.
>
> To further address any potential queries you may have, we have conducted an additional set of ablation experiments on the Rain200L Dataset. For a quicker quantitative assessment, we retrained these models with 100k iterations. Thus, it is noteworthy that this performance is lower compared to the 300k iteration results presented in the paper.
>
> |    | Variants                       | PSNR  | SSIM   |
> |----|------------------|-------|--------|
> | vv1 | $N=2,k_1=3,k_2=3,$ activate $k_1$         | 40.42 | 0.9879 |
> | **Regformer** |$N=2,k_1=3,k_2=5,$ activate $k_1$          | **40.52** | **0.9880** |
> | vv3 | $N=3, k_1=3, k_2=5, k_3=7,$ activate $k_1$| 40.44 | 0.9877   |
> | vv4 | $N=3, k_1=3, k_2=5, k_3=7,$ activate $k_1, k_2$| 40.42 | 0.9876 |
>
> ---
>
> [1] Restormer: Efficient Transformer for High-Resolution Image Restoration. CVPR 2022.
>
> [2] DRSformer: Learning a Sparse Transformer Network for Effective Image Deraining. CVPR 2023.

---

> ### Author Response · Authors · 2023-11-15
> **Response to Reviewer edgT, and thanks for your comments. (PART II)**
>
> Thank you for taking the time to review our paper.
>
> ***Q3:***  One of the most important question in this paper is regarding the novelty. The region-based attention mechanism has been applied by many previous works in various areas. The proposed method does not show the advanced benefits of using the masked attention on the deraining problem, especially the explicit mechanism / design to identify the true features of non-rain regions.
>
> ***Response:*** Thank you for your insightful query regarding the novelty of our approach. You are correct in noting that mask mechanisms have been utilized across various fields. However, our primary contribution lies not in the trivial transfer of the mask mechanism from other fields, but in designing novel and specific mechanisms for the rain removal task. Moreover, we elucidate the corresponding principles behind these mechanisms that can result in SOTA performance.
>
>
> **The specific motivation for using mask for deraining**
>
> Our decision to employ the mask mechanism stems from the unique challenge posed by rain-streaked images. These images require differential enhancement for rain-obscured regions and non-rain areas.
>
> Our mask strategy aims to capture the spatial distribution of such patterns and emphasize valuable features for the recovery of details, providing the network with vital location information to aid in effectively eliminating the rain and restoring image details.
>
> **The novel mask mechanism**
>
> Besides the uniqueness of the motivation, our mask mechanism follows a triple manner, including the formulation of foreground and background masks, as well as their combination, which is different from the mask mechanism in other restoration tasks.
>
> **a) Foreground Mask (rain affected mask):** We utilize the foreground mask to better extract the characteristics of foreground rain streaks. This enables the network to identify the varying rain patterns more effectively, preventing the erroneous removal of rain-like features (as shown in Figure 2, where light spots are mistakenly removed as the rain for the results of DRSformer, IDT, and DualGCN).
>
> **b) Background Mask (rain unaffected mask):** The employment of the background mask aims to emphasize useful features, which can be utilized to recover details in rain streaks (as depicted in Figure 7 in the appendix, where rain streaks in the sky are not separated for the ablation setting without the background mask). This ensures that rain areas can be effectively recovered with desired content via the following "combing mask" stage.
>
> **c) Combining Masks:**
> As shown in Figure 1 of the main paper, a full mask is utilized to incorporate the results from both masks with a transformer block. This technique allows the non-rain areas in the background to guide the restoration of the corresponding rain-streaked foreground areas, resulting in more realistic image details.
>
>
> Besides, **Reviewer ac1q** mentioned that our method first points out the importance of explicitly and individually processing rain-affected (foreground) and unaffected (background) regions.
>
> Moreover, the ablation study (the results, particularly for variants v6, v7, and v8 in Table 3 of the main paper) further validates the superiority of using triple masks.
>
>
>
> Finally, we appreciate your comments and welcome further questions or clarifications.

---

> ### Author Response · Authors · 2023-11-21
> **Waiting for your further feedbacks.**
>
> Dear Reviewer edgT:
>
> We have addressed the questions and concerns raised during the review process and submitted our responses. We would like to know if you have further questions or additional feedback on our revisions. Your comments are invaluable to us.
>
> Best regards,
>
> Authors of paper 1110.

---

### Comment · Area_Chair_eVSy · 2023-12-05
**Comments after rebuttal**

Dear Reviewers,

The manuscript still has divergent ratings.

@All reviewers, do you think that the proposed method is only effective on synthetic datasets and has limited improvement on real-world images?

@Reviewers edgT, ac1q, and FN5q, does the response solve your concerns? Please let me know whether you are in favor of this manuscript or not.

Thanks,

AC

---

### Meta-Review · Area_Chair_eVSy · 2023-12-11

**Metareview:**

This paper introduces a region attention in Transformer to better extract features for image deraining. It received reviews with mixed ratings. The major concerns include the limited novelty, limited performance improvement on real-world images, missing comparisons of running time and so on.

PC/SAC Comment: After calibrating for reviews that are systematically inflated, the review process focused on a careful reading of the  reviews.  The more critical reviews raised valid concerns such as limited novelty and limited evaluations on real-world datasets.  The authors tested on an additional dataset during the rebuttal but the improvement is only 0.03db PSNR, which is not that significant (especially given all the heavy engineering work that went into this method).

**Justification For Why Not Higher Score:**

Although the authors clarify novelty of the paper, Reviewer FN5q still has concerns about the performance of the proposed method.

**Justification For Why Not Lower Score:**

N/A

---

### Decision · Program_Chairs · 2024-01-16

Reject